# SNOV: A Scalable Near-global Optimal Verifier for Neural Networks

## ABSTRACT

Neural networks achieve remarkable performance across domains, yet their deployment in safety-critical settings is limited by robustness concerns. Formal verification offers guarantees but faces a trade-off: complete verifiers scale poorly, while incomplete verifiers either yield loose lower bounds or miss counterexamples due to local optima. We propose a hybrid verifier within a branch-and-bound (BaB) framework that tightens bounds from both sides: an NLP-based upper bound (via complementarity constraints) rapidly rejects unsafe instances, while a relaxation-based lower bound (e.g., $\beta$-CROWN) certifies safe ones. When early stopping is not triggered, the procedure converges to an $\epsilon$-tight interval $[\underline{\ell}, \bar{u}]$ localizing the true optimum $f^\star$. To improve efficiency, we introduce warm-started NLP solves with low-rank KKT updates and a pattern-aligned strong branching strategy that accelerates lower-bound tightening. Experiments on MNIST and CIFAR-10 show that our method (i) produces substantially tighter upper bounds than PGD across perturbation radii, (ii) achieves per-node solves with polynomial-time complexity, and (iii) delivers large end-to-end speedups over MIP-based verification, further amplified by warm-starting, GPU batching, and pattern-aligned branching.

## 1 INTRODUCTION

Although neural networks have achieved remarkable success in learning from and exploiting data, their deployment in safety-critical domains such as power grids has been slow. A key obstacle is the stringent requirement that model outputs satisfy task-specific safety or performance specifications under input perturbations and adversarial attacks. *Neural network verification* (VNN) provides a principled way to certify whether a given model is safe to use in realistic scenarios. Formally, as shown in Figure 1, given a nominal input $x_0$ and a perturbation set $\mathcal{C} = \{x : \|x - x_0\|_p \leq \delta\}$, we encode the speci-

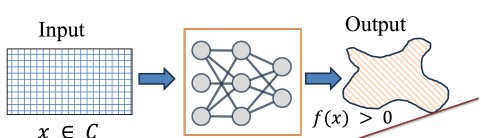

Figure 1: Verification is to decide whether the worst-case output $f^\star = \min_{x \in \mathcal{C}} f(x)$ is nonnegative.

fication as a scalar objective $f : \mathbb{R}^d \to \mathbb{R}$ (e.g., a classification margin or a constraint residual). The instance is *safe* if the global optimal $f^\star \triangleq \min_{x \in \mathcal{C}} f(x) \geq 0$, and *unsafe* if $f^\star < 0$.

Determining whether $f^\star \geq 0$ is notoriously difficult due to the many nonlinear activation functions in modern neural networks making the problem is NP-complete even for piecewise-linear ReLU networks (Katz et al., 2017). Mixed-integer programming (MIP)–based complete verifiers can, in principle, obtain the global optimum, $f^\star$, exactly by encoding each ReLU with binary variables Anderson et al. (2020), but such formulations scale poorly to large networks. Consequently, a large body of work replaces the exact activations with linear or quadratic relaxations (Wong & Kolter, 2018; Zhang et al., 2018a; Wang et al., 2021; Raghunathan et al., 2018) to compute certified lower bounds $\underline{\ell}$ on $f^\star$, and efficiently verify all instances with $\underline{\ell} \geq 0$. These bounds can be tightened further using cutting planes, branching, and related techniques (Zhang et al., 2022a; Yang et al., 2024). However, relaxation-based methods do not directly quantify the optimality gap between the lower bound and the true global optimum, leaving many cases in an "unknown" status. Typically

an additional branch-and-bound (BaB) procedure (Bunel et al., 2020a) is required that splits all unstable neurons (those whose pre-activation bounds straddle zero), and solves exponentially many subproblems.

Rather than computing the lower bounds, another way of verifying the model is to use adversarial attacks, such as projected gradient descent (PGD), to search for counterexamples and obtain upper bounds on $f^\star$ (Madry et al., 2018; Athalye et al., 2018; Zhang et al., 2022b). Unfortunately, these attacks are heuristic and inherently local, and often fail to find counterexamples, becoming trapped in poor local optima far from the global solution. Crucially, tight upper and lower bounds are beneficial for rapidly localizing the potential region of $f^\star$ without incurring excessive branching, and for explicitly obtaining a certified safety margin (the distance between $f^\star$ and the decision boundary) that enables quantitative robustness evaluation of neural networks.

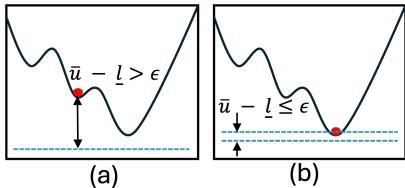

Figure 2: By tightening the lower and upper bounds, branch-and-bound progressively narrows the $\bar{u} - \underline{l}$ from (a) to (b), enabling fast verification without exploring all subproblems.

**Contributions.** To achieve these goals, we define a principled upper bound by solving a nonlinear program (NLP) with complementarity constraints (CC), and couple this with a relaxation-based lower-bounding method (e.g., $\alpha$-CROWN or $\beta$-CROWN within BaB) to obtain a hybrid verifier that shrinks the gap between upper and lower bounds until convergence. The core innovation is a *bidirectional* efficiency link between the primal (upper-bounding) and dual (lower-bounding) components: the NLP solution provides an informative *activation pattern* (which neurons are active/inactive) that we exploit through a pattern-aligned strong branching strategy, tightening lower bounds with substantially fewer branches, while a low-rank warm-start scheme accelerates subsequent NLP solves by updating only a minimally modified model. Specifically, our contributions are:

(i) **NLP–CC upper bounding and $\epsilon$-global verification.** We formulate upper-bound computation as a nonlinear program (NLP) by encoding activation functions exactly via complementarity constraints (CC), denoted by NLP-CC. Any NLP solution yields an upper bound $\bar{u} \geq f^\star$ (the red point in Figure 2). We analyze the robustness of this NLP-based upper bound under large perturbations $\delta$ from a Karush–Kuhn–Tucker (KKT) perspective. By branching on unstable neurons and bounding each subproblem from both sides, we efficiently shrink the gap $\bar{u} - \underline{\ell}$ and localize the candidate region of $f^\star$ (Figure 2), without exhaustively solving all subproblems once the gap is sufficiently small. If $\bar{u} < 0$, we obtain a concrete counterexample (unsafe); if a lower bound $\underline{\ell} > 0$ is certified, the instance is verified safe.

(ii) **Warm-started NLP–CC with low-rank KKT updates.** We accelerate solving NLP–CC within BaB via warm-starting from the previously constructed model. Since each branch only modifies the constraints of a small subset of neurons, we update the KKT system by a low-rank correction and re-solve from this warm start. This preserves accuracy while yielding up to an order-of-magnitude speedup in practice.

(iii) **Pattern-aligned strong branching.** We develop a pattern-aligned strong branching strategy that combines traditional filtered smart branching scores with the NLP-derived activation pattern (active/inactive neurons at the incumbent). Branching scores for unstable neurons are regularized by how well each split aligns with this pattern, leading to fewer, but more informative branches. As shown in Sec. 5.5, this guidance allows $\beta$-CROWN to tighten lower bounds faster than standard filtered smart branching De Palma et al. (2021b).

(iv) **Extensive empirical evaluation.** We conduct an extensive evaluation on MNIST and CIFAR-10 benchmarks, showing that our method (a) yields tighter upper bounds than six bound-propagation baselines and a PGD upper-bounding baseline across perturbation radii, (b) exhibits polynomial running time for obtaining upper bounds, while MIP running-times grow exponentially with the number of binary variables, (c) achieves substantial speedups from NLP warm-starts, and (d) rapidly reduces the lower–upper gap across branch rounds, with pattern-aligned strong branching effectively tightening lower bounds.

## 2 BACKGROUND

**Problem Formulation of Verification** Let $x \in \mathbb{R}^d$ denote the input and $\{(W^{(\ell)}, b^{(\ell)})\}_{\ell=1}^{L}$ the parameters of a feedforward neural network with pre-activations $z^{(\ell)}$ and post-activations $\hat{z}^{(\ell)}$, $f(x)$ is a task-specific *specification* represented by function $f(x)$ such that $f(x) \geq 0$ means the property holds. One of the common specifications is a linear function of the last layer $z^L$. Without losing generalization, we employ the target-specific to verify the output, i.e., $f(x) = z_k^L - z_a^L$, where $k, a$ are the label and attacked class respectively. We consider inputs constrained to an $l_p$ ball $\mathcal{C} = \{x : \|x - x_0\|_p \leq \delta\}$ (most commonly $p \in \{2, \infty\}$). Let $\mathcal{C}$ denote the chosen input set. Verification asks for the minimum specification value over the admissible inputs: $\min_{x \in \mathcal{C}} \ f(x), \text{s.t.} \quad \hat{z}^{(0)} = x, z^{(\ell)} = W^{(\ell)}\hat{z}^{(\ell-1)} + b^{(\ell)}, \ \ell = 1, \ldots, L, \hat{z}^{(k)} = \sigma(z^{(k)}), \ k = 1, \ldots, L-1.$, where $\sigma$ represents the ReLU activation function. If the optimal value $f^*$ is nonnegative, the specification holds for all $x \in \mathcal{C}$. Note that it is beneficial to first compute the pre-activation bounds $[\ell^\ell, u^\ell]$ using fast bound-propagation methods like CROWN (Zhang et al. (2018a)) before thoroughly searching for $f^*$. Though mixed-integer programming (MIP) can directly compute the global optimum, it scales poorly to large networks. Here, we introduce alternative formulations that address the same problem more efficiently, namely nonlinear programming (NLP) with complementarity constraints and bound-propagation–based methods, along with related concepts.

**Nonlinear Programming with Complementarity Constraints (NLP-CC).** Representing ReLU activations with *equivalent* complementarity constraints in a nonlinear program (NLP) has recently been studied in the optimization and control literature Yang et al. (2022) and applied to neural network verification Chehade et al. (2025). Consider layer $\ell$ with pre-activation $z^{(\ell)}$ and post-activation $\hat{z}^{(\ell)} = \text{ReLU}(z^{(\ell)})$. Then, introduce nonnegative variables $p^{(\ell)}, q^{(\ell)}$ and impose

$$z^{(\ell)} = p^{(\ell)} - q^{(\ell)}, \qquad p^{(\ell)} \geq 0, \ q^{(\ell)} \geq 0, \qquad p^{(\ell)} \odot q^{(\ell)} = 0, \tag{1}$$

which encodes the ReLU operator exactly. For numerical robustness, it is suggested to use a softened form $p^{(\ell)} \odot q^{(\ell)} \leq \varepsilon_{\text{comp}} \mathbf{1}$ with $\varepsilon_{\text{comp}} \in [10^{-8}, 10^{-5}]$, whose impact on the upper bound $\bar{u}$ is negligible (below $10^{-5}$) in the same spirit as NLP solver tolerances. Note we use constraints of intermediate bounds $[l^\ell, u^\ell]$ from cheap bound propagation methods like $\alpha-$CROWN. This *activation-exact* formulation extends directly to CNN and ResNet architectures.

**Bound Propagation.** A broad family of incomplete verifiers is based on bound propagation, which propagates linear upper and lower bounds through the network (forward or backward) to obtain a certified lower bound on $f^*$. A representative method is $\alpha$-CROWN Zhang et al. (2018a), which computes a linear lower-bound function $\min_{x \in \mathcal{C}} f_{\alpha\text{-CROWN}}(x) = a_{\alpha\text{-CROWN}}^\top x + c_{\alpha\text{-CROWN}} \leq f^*$, where $a_{\alpha\text{-CROWN}}$ and $c_{\alpha\text{-CROWN}}$ are obtained from the network parameters and input bounds. $\beta$-CROWN follows a similar convex-relaxation principle, but explicitly incorporates the split constraints $\mathcal{Z}$ Wang et al. (2021). It computes a lower bound for the subproblem $\min_{x \in \mathcal{C} \cap \mathcal{Z}} f(x)$ by optimizing a linear surrogate objective $g(\beta)$ that depends on an auxiliary variable $\beta$.

**Branch and Bound.** Branch-and-bound (BaB) is a standard tool for tightening bounds by iteratively splitting unstable neurons and is widely used in complete verifiers to ensure completeness. Branching on an unstable neuron $z_j^{(\ell)}$ creates two subdomains, $\mathcal{C}_{\ell,j} = \{\mathcal{C} \cap \mathcal{C}_{z_j^{(\ell)} \geq 0}\}$ and $\mathcal{C}_{\ell,j} = \{\mathcal{C} \cap \mathcal{C}_{z_j^{(\ell)} < 0}\}$, while the bounding step computes lower bounds $\underline{l}_i$ for each subdomain $\mathcal{C}_i$. A subdomain is certified safe if $\underline{l}_i > 0$. Most existing branching rules are heuristic, including Filtered Smart Branching (FSB) De Palma et al. (2021b) and BaBSB Bunel et al. (2020b), which choose splits based on estimated improvements in the objective bound. FSB building upon BaBSB operates in two stages: (i) it assigns a cheap score to each unstable neuron, approximating the lower-bound gain, and retains a small candidate set $D_{\text{FSB}} \subset (k, j)$; (ii) for each candidate, it evaluates a fast dual lower bound Dvijotham et al. (2018) to compute a strong-branching score $s(\mathcal{C}_i)$ for the subdomain $\mathcal{C}_i$, and then branches on the neuron with the largest $s(\mathcal{C}_i)$.

## 3 RELATED WORK

Verification methods are typically categorized as *complete* or *incomplete* based on their ability to certify all input cases. Complete verifiers either certify robustness or find counterexamples with formal guarantees for every instance, while incomplete verifiers handle only a subset, leaving the rest undecided. While our method formally belongs to the class of incomplete verifiers, it successfully handles nearly all instances in practice and tightly localizes the global optimum within a small region.

**Complete Verifiers.** Complete verifiers typically combine a global search procedure with relaxed or exact activation encodings to examine (in principle) all activation patterns. One major line of work (e.g., MIP, $\beta$-CROWN-based BaB, MN-BaB with multi-neuron relaxations, SDP-BaB) couples convex-relaxation-based lower bounds with branch-and-bound, ultimately splitting on unstable neurons until every relevant pattern is resolved Bunel et al. (2018); De Palma et al. (2021c); Xu et al. (2021); Henriksen & Lomuscio (2020); Wang et al. (2021); De Palma et al. (2021a); Lu & Kumar (2019); Ferrari et al. (2022). Abstraction–refinement tools such as nnenum and certain NNV/CORA modes instead search over abstract elements Bak et al. (2021); Müller et al. (2022), adaptively refining zonotope or star-set representations until the over-approximate reachable set is either separated from the unsafe region or a violating state is found. A third line, including SMT- and CDCL(T)-based systems such as Marabou, encodes ReLU phases and other piecewise-linear structure as Boolean variables and performs DPLL/CDCL search with learned clauses, which can be viewed as a symbolic branch-and-bound over activation patterns Katz et al. (2019); Scheibler et al. (2015); Duong et al. (2023); Liu et al. (2024); Biere et al. (2009); Marques-Silva et al. (2021). While complete methods guarantee sound and exhaustive results, they scale poorly to large networks due to the NP-complete nature of exact verification. In contrast, we treat verification as an optimization problem, bounding the global optimum within a provable $\epsilon$-gap. This avoids enumerating all activation patterns when the gap is small. Empirically, our method significantly reduces runtime with only minor loss in verification coverage, benefiting from structural properties of trained networks, such as sparsity and dominant patterns.

**Incomplete Verifiers.** Incomplete verifiers rely either on relaxation-based bounds (e.g., $\alpha$-CROWN, DeepPoly, zonotope and PRIMA without BaB) Wong & Kolter (2018); Zhang et al. (2018b); Singh et al. (2019); Müller et al. (2021); Salman et al. (2019) or heuristic searches such as gradient-based attacks and randomized smoothing Madry et al. (2018); Cohen et al. (2019); Tong et al. (2024). While they can return certified lower bounds or concrete counterexamples, they do not explore the full nonconvex domain and may fail to verify due to relaxation error or suboptimal local minima; moreover, the gap to the true global optimum is unknown. We instead bound the global optimum $f^\star$ from both sides and stop early when the upper bound becomes negative (yielding a counterexample) or the lower bound becomes positive (certifying robustness). More generally, every result is returned with an interval $[\underline{\ell}, \bar{u}]$ of width at most $\epsilon$, providing either a counterexample, a robustness certificate, or an $\epsilon$-tight optimality gap.

## 4 SNOV: A SCALABLE NEAR-GLOBAL OPTIMAL VERIFIER

*A key feature of our verifier, SNOV, is its bidirectional efficiency: upper bounding enables rapid rejection of unsafe cases, while lower bounding dominates the effort in certifying safe ones. Even when early stopping is not triggered, the final converged upper and lower bounds are guaranteed to be tight, localizing $f^\star$ within a small region. This yields an accurate safety margin, defined as the distance from $f^\star$ to the decision boundary (e.g., zero).*

### 4.1 HIGH-LEVEL STRUCTURE.

SNOV is a hybrid verifier that couples an *activation-exact* local NLP-CC solver (for tight *upper* bounds) with $\beta$-CROWN bound propagation (for certified *lower* bounds) within a branch-and-bound (BaB) loop (Fig. 3). For the $i$-th subdomain $\mathcal{C}_i$ (a partial assignment of ReLU states), SNOV computes the lower bound $\underline{l}(\mathcal{C}_i)$ via $\beta$-CROWN and the upper bound $\bar{u}(\mathcal{C}_i)$ via activation-exact NLP. The loop terminates when the gap $\bar{u} - \underline{l} \leq \epsilon$, yielding an $\epsilon$-optimal cer-

tificate; if the global lower bound satisfies $\underline{l} > 0$, the network is certified *Safe*, while any subdomain with $\bar{u} < 0$ is immediately declared *Unsafe* (early stop), as shown in Figure 4. To enhance efficiency on both sides, SNOV employs a lightweight upper-bounding procedure and a pattern-aligned strong branching strategy. Taking into account the asymmetric runtime and information content of the lower and upper bounds, we further introduce coordination mechanisms between them to ensure both accuracy and fast convergence. The overall procedure is summarized in Algorithm 1, with step-by-step details provided in Appendix E.

### 4.2 COMPONENT I: NLP-CC PRODUCES A SOUND UPPER BOUND

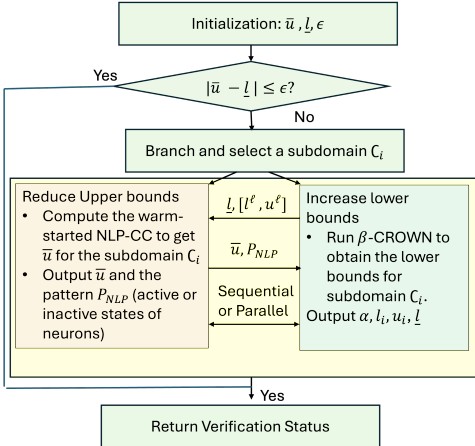

Figure 3: **Flowchart of SNOV with bidirectional flow.** $\beta$-CROWN supplies lower bounds to NLP-CC, and NLP-CC returns upper bounds and activation patterns that guide pruning and branching.

**Exact Reformulation and Upper bound** Though many smooth nonlinear approximations of ReLU exist, we adopt a complementarity-constraint (CC) formulation because it is exactly equivalent to the original ReLU problem, ensuring that any solution of the resulting NLP–CC yields a sound upper bound for the true verification objective.

**Proposition 1** (Soundness of the NLP–CC Upper Bound)**.** *Consider a ReLU network with specification function $f$ and perturbation set $\mathcal{C}$, and denote the optimal value of the original verification problem by $f^{\star} = \min_{x \in \mathcal{C}} f(x)$. Let $(x, \dots)$ be any feasible point of the nonlinear program with complementarity constraints (NLP–CC) obtained from the exact reformulation of the ReLU network. Then $f(x) \geq f^{\star}$. In other words, every feasible solution of the NLP–CC provides a valid upper bound on the true optimum $f^{\star}$.*

We present the core principles here and give a simple two-layer network example in Appendix D to illustrate the exact complementarity-based reformulation of ReLU networks. A ReLU activation can be written as the solution of a projection QP: $\mathrm{ReLU}(z) = \arg\min_{\hat{z} \geq 0} \frac{1}{2} \|\hat{z} - z\|_2^2$. Since this QP is convex, its optimal solution is characterized exactly by the Karush–Kuhn–Tucker (KKT) conditions. Let $p := \hat{z}$ be the primal variable and $q \geq 0$ the dual variable associated with the constraint $p \geq 0$. The Lagrangian is $\mathcal{L}(p, q) = \frac{1}{2} \|p - z\|_2^2 - q^\top p$, and the KKT conditions are $p \geq 0, \quad q \geq 0, \quad p - z + q = 0, \quad p \odot q = 0$. These complementarity conditions *exactly* encode $p = \mathrm{ReLU}(z)$, i.e., there is no approximation. Embedding these KKT conditions neuronwise for all layers yields an NLP with complementarity constraints (NLP-CC) that is *equivalent* to the original ReLU network mapping: for all inputs $x$, $f_{\mathrm{NLP}}(x) = f(x)$. Therefore, solving the verification problem $\min_{x \in \mathcal{C}} f_{\mathrm{NLP}}(x)$ via NLP-CC produces a feasible (possibly local) solution whose objective value $f_{\mathrm{NLP}}^{\star}$ satisfies $f^{\star} \leq f_{\mathrm{NLP}}^{\star}$, providing a sound upper bound on the true global optimum $f^{\star}$ of the original problem.

**Warm-start with Low-rank KKT Updates.** The efficiency of solving each NLP-CC subproblem depends strongly on the initialization. We warm-start the interior-point method solver by reusing the primal and dual solution from the parent node and applying only small, localized modifications to the model. The NLP-CC is solved using a primal–dual interior-point method (e.g., IPOPT from Wächter & Biegler (2006)), which follows a sequence of perturbed KKT systems, while maintaining strict feasibility of slacks and multipliers. The detailed explanation of primal–dual interior-point is in Appendix B. When branching on an unstable ReLU unit, only a few neurons change phase. For each such neuron $(k, j)$, we update its bounds and replace the complementarity surrogate with the exact linear relations of the chosen

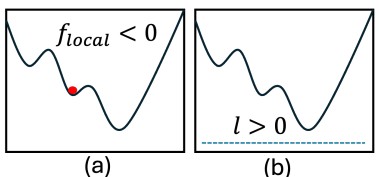

Figure 4: Early stopping rules that cut search cost.

phase: *active:* $q_j^k = 0$, $p_j^k = z_j^k$, $l_j^k = 0$; *inactive:* $p_j^k = 0$, $z_j^k = -q_j^k \leq 0$, $u_j^k = 0$. This modification removes one complementarity constraint and introduces up to three linear equalities, thereby altering only a small block of the global KKT system. Consequently, the change in the KKT matrix is a low-rank update (rank $\leq 4$). Because the remainder of the model and dual information is preserved, the warm-started iterate lies close to the new path, and the interior-point method typically converges within only a few Newton steps.

**Early stopping and counterexample.** If the NLP returns $\bar{u}(d) < 0$, a concrete counterexample is found, and the verification status of the network is labeled as*Unsafe*.

### 4.3 COMPONENT II: PATTERN-ALIGNED STRONG BRANCHING FOR PRINCIPLE-GUIDED LOWER BOUNDING

Although various lower-bounding methods exist, selecting and leveraging an appropriate one is non-trivial—especially when upper-bound information is available. We summarize three core principles guiding our design.

**Principle (1): Global characterization and early stopping of lower bounds.** The lower-bounding method must provide a global perspective to ensure meaningful comparison with the upper bound. To be compatible with the BaB framework, it must also adapt to domain splits and progressively tighten bounds. $\beta$-CROWN is a suitable candidate, as it computes tight lower bounds for each subdomain, as described in Section 2, and supports GPU acceleration with batches. SNOV triggers early stopping and reports a verified-safe result as soon as all active lower bounds satisfy $\underline{l} \geq 0$, unless the user explicitly requests the exact optimal gap.

**Principle (2): Pattern-aligned strong branching.** Branching is crucial for efficiency: identifying early the subdomain containing $f^\star$ significantly reduces the number of subproblems to solve. The goal is not to reduce the total number of branches, but to prioritize the branch most likely to lead to the worst-case scenario. Moreover, after branching, it is important to order subdomains $\mathcal{C}_i$ to explore first those closer to $f^\star$. For domain selection, we follow the conventional rule based on the lower bounds $\underline{l}_i$ Wang et al. (2021).

**Pattern-aligned strong branching strategy.** To further improve branching, we introduce a pattern-aligned strong branching strategy built on FSB. As shown in Section 4.2, the local optimum $f_{\text{NLP}}$ obtained from the NLP-CC subproblem often lies close to $f^\star$. Thus, the activation pattern $p_{\text{NLP}}$ associated with $f_{\text{NLP}}$ can indicate the region containing $f^\star$. We define a pattern-aligned score $s_p(\mathcal{C}_i)$ for the $i$-th domain as

$$s_p(\mathcal{C}_i) = s(\mathcal{C}_i) + \lambda \cdot m\big(p(\mathcal{C}_i), p_{\text{NLP}}\big), \tag{2}$$

where $s(\mathcal{C}_i)$ is the strong-branching score from FSB, and $m(p(\mathcal{C}_i), p_{\text{NLP}}) = \frac{n_m}{|\mathcal{U}|}$ measures the fraction of unstable neurons in $\mathcal{U}$ whose phases match between domain $\mathcal{C}_i$ and $p_{\text{NLP}}$. Here, $n_m$ is the number of matched neurons, and $\lambda > 0$ controls the influence of pattern alignment.

### 4.4 COMPONENT III: BaB ORCHESTRATION

We maintain a queue of domains $\mathcal{C}_i$. At each iteration: (1) Compute $\underline{l}(\mathcal{C}_i)$ via $\beta$-CROWN; prune if $\underline{l}(\mathcal{C}_i) > 0$. (2) Due to the asynchronous running time, we re-solve the NLP only every $\tau_{\max}$ iterations, using warm-start and low-rank updates; we recommend setting $\tau_{\max}$ to roughly match the number of $\beta$-CROWN bound computations that take the same time as one NLP solve, and update $\bar{u}(\mathcal{C}_i)$; prune if $\bar{u}(\mathcal{C}_i) < 0$. (3) If $\bar{u} - \underline{l} \leq \epsilon$ or exceed the maximum iterations $t_{\max}$, terminate with an $\epsilon$-certificate. (4) Otherwise, *branch* on the most impactful unstable neuron using a strong-branching score that mixes (a) predicted bound tightening from $\beta$-CROWN and (b) patterns from the NLP.

**Scalability notes.** (1) Activation-exact NLP solves reuse KKT factorizations with rank-$\leq 4$ updates. (2) $\beta$-CROWN runs in batched mode across nodes. (3) Early stopping ($\bar{u} < 0$ or $\underline{l} > 0$) curtails deep trees. Together these yield fast per-node times and robust performance at large perturbation radii.

---

**Algorithm 1** SNOV: Scalable Near Optimal Verification

---

1: **Input and Initialization:** $\{W^i, b^i\}_{i=1}^L, \mathcal{C}, \epsilon; t = 0, \tau = 0, \tau_{\max};$
2: **Root bounds:** run $\alpha$-CROWN to compute intermediate bounds $[l^\ell, u^\ell]$ and global lower bound$\underline{l}$; if $\underline{l} > 0$, **return** Safe
3: **Root incumbent:** Solve NLP-CC to obtain $\bar{u}$; if $\bar{u} < 0$, **return** Unsafe
4: Set the domain set $\mathcal{D} = \{\underline{l}, \bar{u}, \mathcal{C}\}$.
5: **while** $\bar{u} - \underline{l} > \epsilon$ **and** $\mathcal{D} \neq \emptyset$ **and** $t \leq t_{\max}$ **do**
6:     Branch on an unstable neuron using the pattern-aligned strong-branching score (equation 2) and pop $\mathcal{C}_i$ the subdomain from $\mathcal{D}$.
7:     **for** $h \in \{0, 1\}$ **do**                          ▷ split into inactive/active child domains
8:         Apply the split to obtain $\mathcal{C}_i^{(h)}$, and run $\beta$-CROWN to compute $\underline{l}(\mathcal{C}_i^{(h)})$; increment $\tau \leftarrow \tau + 1$
9:         **if** $\tau > \tau_{\max}$ **then**
10:            Update the KKT system and solve the warm-started NLP to get $\bar{u}(\mathcal{C}_i^{(h)})$; reset $\tau \leftarrow 0$
11:        **end if**
12:        **Prune** if $\bar{u}(\mathcal{C}_i^{(h)}) < 0$ or $\underline{l}(\mathcal{C}_i^{(h)}) > 0$; otherwise, push $\mathcal{C}_i^{(h)}$ to the domain set $\mathcal{D}$
13:    **end for**
14:    Update global bounds: $\bar{u} \leftarrow \min\{\bar{u}, \bar{u}(\mathcal{C}_i^{(0)}), \bar{u}(\mathcal{C}_i^{(1)})\}, \underline{l} \leftarrow \max\{\underline{l}, \underline{l}(\mathcal{C}_i^{(0)}), \underline{l}(\mathcal{C}_i^{(1)})\}$
15: **end while**
16: **return** $\epsilon$-optimal certificate: Safe if $\underline{l} > 0$, Unsafe if $\bar{u} < 0$, otherwise the gap $\bar{u} - \underline{l}$.

---

## 5 NUMERICAL RESULTS

**Experimental Setup.** We evaluate our method on MNIST LeCun et al. (2010) (input dimension 784) and CIFAR-10 Krizhevsky et al. (2009) (input dimension 3,072)) under varying perturbation magnitudes. The model with MNIST achieves an averaged 97% classification accuracy even with raw datasets without normalization, while the model with CIFAR10 only produces averaged 57.03% accuracy with normalization, fine-tuned learning rates, and cosine warmup scheduler Wolf et al. (2020). We compare against seven baselines—IBP, CROWN-IBP, CROWN, $\alpha$-CROWN, MIP, $\alpha$-$\beta$-CROWN, and PGD—focusing on bound tightness and runtime across scenarios. Nonlinear programs and mixed-integer formulations are implemented in Pyomo Hart et al. (2024) and solved using IPOPT Wächter & Biegler (2006) (for NLP) and Gurobi (for MIP). Note that we first employ $\alpha$-CROWN to obtain the intermediate bounds before solving the MIP as recommended in literature. Experiments are conducted on two systems: (i) a Mac with Apple M4 (10-core CPU/GPU) for Sec.5.1-5.3 and (ii) a server with 32-core AMD CPU and 64 GPUs for Sec. 5.4 and 5.5. We first evaluate SNOV$_u$ in isolation—examining its tightness, efficiency, and warm-start effectiveness—against all baselines through system (i). We then present full-system results where SNOV$_u$ is combined with $\beta$-CROWN and our pattern-aligned strong branching through the system (ii). Implementation details are provided in Appendix A. All code and datasets will be publicly released on GitHub https://github.com/SNOV2025/SNOV.

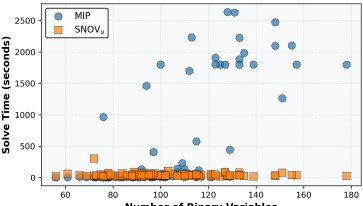

**Metrics.** To quantify verifier sensitivity under perturbations of radius $\delta$, we report the absolute and relative relaxation errors of certified lower bounds: $\Delta_\delta = \left|\bar{u} - f^\star\right|$ or $\left|\underline{l} - f^\star\right|$, $\bar{\Delta}_\delta = \Delta_\delta / \left|f^\star\right|$, $f^\star$ is the ground-truth minimum obtained by an exact or high-fidelity solver. *Upper bounding rate* is defined as the ratio of obtaining optimal upper bounds and the total number of cases.

Figure 5: Runtime distribution over 100 CIFAR-10 cases for MIP and SNOV$_u$.

### 5.1 TIGHTNESS COMPARISON OF SNOV$_u$

We first compare the average lower bounds obtained by bound-propagation methods (IBP, CROWN-IBP, CROWN, $\alpha$-

Table 1: Averaged Performance of Ten Images of **MNIST** Dataset Under **Large** Perturbations $\|\delta\|_\infty \leq 0.1$

| Method | IBP | CROWN-IBP | CROWN | $\alpha$-CROWN | $\mathbf{SNOV}_u$ | MIP |
|---|---|---|---|---|---|---|
| $\Delta_{0.1}$ | 128.6 | 68.01 | 42.73 | 35.53 | **0.43** | 0 |
| $\bar{\Delta}_{0.1}$ | 18.85 | 9.69 | 5.75 | 5.02 | **0.039** | 0 |
| Time (s) | 0.0011 | 0.0029 | 0.0022 | 0.1119 | 1.3624 | 161.6231 |

Table 2: Averaged Performance of Ten Images of **MNIST** Dataset Under **Small** Perturbations $\|\delta\|_\infty \leq 0.01$

| Method | IBP | CROWN-IBP | CROWN | $\alpha$-CROWN | $\mathrm{SNOV}_u$ | MIP |
|---|---|---|---|---|---|---|
| $\Delta_{0.01}$ | 16.2651 | 2.57 | 0.4719 | 0.4673 | **0.0004** | 0 |
| $\bar{\Delta}_{0.01}$ | 2.871 | 0.3825 | 0.0689 | 0.0681 | **$4.04e^{-05}$** | 0 |
| Time (s) | 0.0006 | 0.0012 | 0.0014 | 0.096 | 0.3808 | 0.2364 |

| | PGD | $\mathrm{SNOV}_u$ |
|---|---|---|
| Upper Bounding Rate (%) | 42 | **100** |
| $\Delta_{0.03}$ | 0.204 | **0.003** |
| $\bar{\Delta}_{0.03}$ | 5.781 | **0.008** |

| | PGD | $\mathrm{SNOV}_u$ |
|---|---|---|
| Upper Bounding Rate (%) | 21 | **100** |
| $\Delta_{0.01}$ | 0.17 | **0.0005** |
| $\bar{\Delta}_{0.01}$ | 0.32 | **0.0009** |

Table 3: Comparison using CIFAR10 with $\delta = 0.03$

Table 4: Comparison using CIFAR10 with $\delta = 0.01$

CROWN), our NLP-based upper bound $\mathrm{SNOV}_u$, and the MIP solver on 10 MNIST cases under large ($\|\delta\|_\infty \leq 0.1$) and small ($\|\delta\|_\infty \leq 0.01$) perturbations. Bound-propagation methods are fast but produce very loose bounds, with $\bar{\Delta}_{0.1}$ values ranging roughly from 0.0681 to 18.85 times. As shown in Tables 1 and 2, the relative (or absolute) optimality gaps of bound-propagation methods grow substantially as the perturbation radius increases from 0.01 to 0.1. We also report verification rates: for small perturbations (Table 2), looser methods such as IBP and CROWN-IBP fail to verify all cases (verification rates of 10% and 90%, respectively), whereas $\mathrm{SNOV}_u$ successfully certifies every instance. In contrast, $\mathrm{SNOV}_u$ consistently maintains a gap below 0.05, yielding tight bounds in both the $f^\star > 0$ and $f^\star < 0$ cases. Moreover, when $\delta$ is small, MIP can be relatively fast, but its cost increases dramatically under larger perturbations (e.g., Table 1 shows an average of 161 seconds). This is primarily due to the rapid growth in the number of binary variables as $\delta$ increases—a phenomenon we analyze in detail with additional statistics in the next section.

Next, we compare our upper bounds with those obtained by PGD on the first 100 CIFAR-10 cases for $\delta = 0.01$ and $\delta = 0.03$ (Table 3 and 4). PGD efficiently finds adversarial examples (averaging 0.002 seconds), but the resulting upper bounds are very loose, with optimality gaps often exceeding 0.2. Moreover, PGD fails to produce valid upper bounds whenever no counterexample is found, leading to a low upper-bounding rate. In contrast, $\mathrm{SNOV}_u$ consistently yields upper bounds with gaps below 0.005 and successfully returns bounds for all 100 cases.

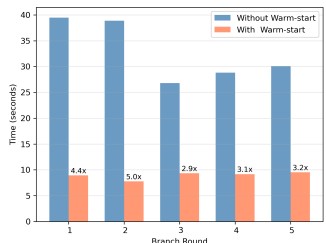

Figure 6: Speedup with warm up

## 5.2 Efficiency Comparison of $\mathrm{SNOV}_u$

We note that $\mathrm{SNOV}_u$ is not always faster than MIP; thus, we analyze when $\mathrm{SNOV}_u$ provides the most benefit. Figure 4 reports the runtimes of 100 CIFAR-10 cases solved by MIP and $\mathrm{SNOV}_u$. The runtime is strongly correlated with the number of binary variables, which grows with network size and perturbation radius. When the number of binary variables is below

roughly 180, $\text{SNOV}_u$ completes within 0–30 seconds, whereas MIP runtimes increase almost exponentially—from several millionseconds to hundred seconds and even beyond the time limit.

Many MIP runs exceed the 2500-second cap and are recorded at this maximum. Once the number of binary variables exceeds about 120, MIP is typically two orders of magnitude slower than $\text{SNOV}_u$, highlighting a clear efficiency advantage. These results suggest that SNOV is particularly recommended for cases with at least a few hundred binary variables, arising from larger networks or larger perturbations.

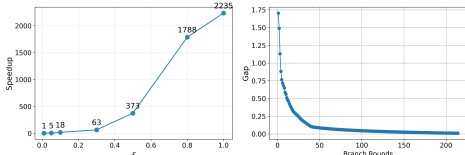

Figure 7: (a) Speedup with $\epsilon \in [0, 1]$ and (b) Gaps reduction with the number of branch rounds

### 5.3 EFFICIENCY IMPROVEMENT OF $\text{SNOV}_u$ VIA WARM-UP

We compare the runtime of $\text{SNOV}_u$ when solved from scratch versus using warm-starts with low-rank KKT updates. Using a representative case, we report the runtime over branch rounds 1–5. As shown in Figure 6, warm-starting substantially accelerates each NLP-CC solve, yielding a 2–5× speedup. We update the upper bound only when the newly solved value is lower; otherwise, the previous upper bound is retained.

### 5.4 PERFORMANCE OF SNOV WITHIN BAB

We examine a challenging instance (case 42), where MIP takes over 2000 seconds to solve for $f^\star$, to illustrate the performance of SNOV. The right one of Figure 7 shows how the optimality gap shrinks with increasing branch rounds. The gap decreases rapidly as more unstable neurons are split, primarily due to the rise in lower bounds—since $\text{SNOV}_u$ already finds an upper bound close to $f^\star$ in early iterations. Since total runtime depends on the target optimality gap $\epsilon$, we report the speedup of SNOV over MIP across $\epsilon \in [0.01, 0.05, 0.1, 0.3, 0.5, 0.8, 1]$ in the left one of Figure 7. The results show that speedup grows rapidly with increasing $\epsilon$, emphasizing the importance of selecting an appropriate tolerance to balance solution quality and efficiency.

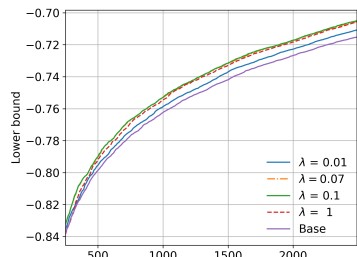

Figure 8: Lower bounds rise faster with an nonzero $\lambda \in [0, 1]$, where "base" denotes $\lambda = 0$, i.e., the standard FSB method.

### 5.5 PERFORMANCE OF PATTERN-ALIGNED STRONG BRANCHING

We use the same instance as in the previous section to evaluate the effect of the pattern-aligned strong branching strategy, which is designed to accelerate lower bound computation. Figure 8 shows that, across different values of $\lambda$, our method increases the lower bound more rapidly than the baseline (standard FSB) as branch rounds progress. The improvements become pronounced after approximately 200 rounds and stabilize after 500 rounds, with $\lambda = 0.1$ yielding the best performance.

## 6 CONCLUSIONS AND FUTURE WORK

Neural network verification is NP-complete, and fully complete methods are often prohibitively expensive in practice. This work targets $\epsilon$-global optimality by jointly tightening upper and lower bounds via a hybrid scheme: an NLP-based upper bound using an exact reformulation of nonconvex activations, combined with $\beta$-CROWN lower bounds within a BaB framework. We demonstrate two–to–three orders of magnitude speedup over MIP-based verifiers, depending on $\epsilon$. To further improve scalability, we introduce a warm-start strategy with low-rank KKT updates that reduces NLP re-solving cost, and a pattern-aligned strong branching rule that systematically strengthens lower bounds across branch rounds. Future work will more fully explore the interaction between advanced local solvers and convex relaxations to design scalable, near–globally optimal verification pipelines, with the long-term goal of certifying large-scale networks, including (near) real-time settings.

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

# A  APPENDIX

**APPENDIX A**: EXPERIMENTAL DETAILS

**Settings and hyperparameters**  We implement a rank-4 warm start for the NLP subproblems following IPOPT's warm-start guidelines Wächter & Biegler (2006). In Pyomo, warm starts are realized by attaching the dual, `ipopt_zL_in`, and `ipopt_zU_in` suffixes to the model, which import/export constraint duals and variable-bound multipliers between solves. Before each re-solve we preload the primal variables and these multipliers and invoke IPOPT with `warm_start_init_point=yes`, which accelerates convergence on closely related subproblems.

Results on upper-bounding performance (Secs. 5.1–5.3) are obtained on a single Mac machine (Apple M4 CPU/GPU), while the overall BaB experiments (Secs. 5.4–5.5) are run on a single GPU of the server (we report the exact CPU/GPU models there).

**Network architecture for MNIST.** We consider a fully connected feedforward network for MNIST dataset, denoted `NoSoftmaxNet`, with two hidden layers of width $h = 50$, $d = 784$, and a linear output layer producing $K = 10$ logits. The architecture is summarized in Table 5.

Table 5: Architecture of the `NoSoftmaxNet` model.

| Layer | Type | Input dimension | Output dimension |
|:-----:|:----:|:---------------:|:----------------:|
| 1 | Linear+ReLU | 784 | 50 |
| 2 | Linear+ReLU | 50 | 50 |
| 3 | Linear | 50 | 10 |

**Training procedure for MNIST.** We train the network using mini-batch stochastic optimization with the Adadelta optimizer and a step-wise learning-rate schedule. Specifically, given training data $\{(x_i, y_i)\}$ with images $x_i \in \mathbb{R}^{28 \times 28}$ and labels $y_i \in \{0, \ldots, 9\}$, we first flatten each image to a vector in $\mathbb{R}^{784}$ and feed it to the network $f_\theta(\cdot)$ with parameters $\theta$. For each mini-batch $\mathcal{B}$, the model outputs logits $f_\theta(x_i)$, and we minimize the cross-entropy loss

$$\mathcal{L}(\theta; \mathcal{B}) = \frac{1}{|\mathcal{B}|} \sum_{(x_i, y_i) \in \mathcal{B}} \ell_{\mathrm{CE}}\big(f_\theta(x_i), y_i\big),$$

where $\ell_{\mathrm{CE}}$ denotes the standard multi-class cross-entropy. Gradients $\nabla_\theta \mathcal{L}$ are computed by backpropagation, and the parameters are updated by Adadelta with initial learning rate `lr`. A StepLR scheduler is applied to the optimizer, reducing the learning rate by a factor `gamma` every `step_size` epochs. After training, the final model parameters are stored using `torch.save`.

**Network Architecture for CIFAR10** We use a fully connected ReLU network with two hidden layers.

Table 6: Feedforward network architecture (fully connected ReLU MLP).

| Layer | Type | Shape / Output Dim | Activation |
|:-----:|:----:|:------------------:|:----------:|
| Input | Flatten | $32 \times 32 \times 3 \to 3072$ | – |
| Hidden 1 | Linear | $3072 \to 256$ | ReLU |
| Hidden 2 | Linear | $256 \to 256$ | ReLU |
| Output | Linear | $256 \to 10$ | – |

**Training procedure for CIFAR10.** We train the model with a cross-entropy loss enhanced by label smoothing and optimize the parameters using AdamW with weight decay and fixed $(\beta_1, \beta_2) = (0.9, 0.999)$. A cosine learning-rate schedule with a warm-up phase gradually increases the learning rate during the first few epochs, then decays it. During each epoch, we iterate over mini-batches. After each backward pass, gradients are clipped to a maximum norm of 1.0 before performing an AdamW update.

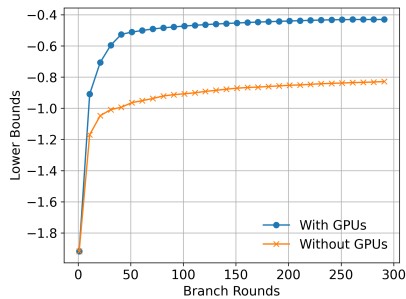

Figure 9: Lower bounds improvement with and without GPU in bath mode

**APPENDIX**
**B**: INTERIOR-POINT-METHOD FOR SOLVING NLP-CC

**Basic NLP and KKT Conditions** Consider a general nonlinear program (NLP):

$$\min_{x \in \mathbb{R}^n} \quad f(x)$$

$$\text{s.t.} \quad h(x) = 0 \quad \text{(equalities)}$$

$$\qquad g(x) \geq 0 \quad \text{(inequalities)}$$

Let $x$ = all variables (input, $z$, $\hat{z}$, $p$, $q$, ...), $h(x) = 0$ represents all equality constraints (e.g., network affine relations, phase constraints, etc.), $g(x) \geq 0$ includes inequality constraints (e.g., bounds, complementarity, etc.).

The Karush-Kuhn-Tucker (KKT) conditions (first-order optimality) are:

$$
\begin{aligned}
\nabla_x L(x, \lambda, \mu) &= 0 &&\text{(stationarity)} \\
h(x) &= 0 &&\text{(primal feasibility)} \\
g(x) &\geq 0, \quad \mu \geq 0 \\
\mu_i g_i(x) &= 0 &&\forall i \quad \text{(complementarity)}
\end{aligned}
$$

where the Lagrangian is defined as:

$$
L(x, \lambda, \mu) = f(x) + \lambda^\top h(x) - \mu^\top g(x)
$$

Interior-point methods aim to **drive the system to a KKT point** while staying *inside* the feasible region.

**Primal-Dual Interior Point (IPM)** Modern solvers (e.g., IPOPT) implement a **primal-dual** interior point method (IPM). First, reformulate the inequality constraints using slack variables:

$$
g(x) \geq 0 \quad \Longleftrightarrow \quad g(x) - s = 0, \quad s \geq 0
$$

with slack variables $s \in \mathbb{R}^m$. Then the KKT conditions become:

$$
\begin{aligned}
\nabla_x L(x, \lambda, \mu) &= 0 \\
h(x) &= 0 \\
g(x) - s &= 0 \\
s &\geq 0, \quad \mu \geq 0 \\
s_i \mu_i &= 0, \quad \forall i
\end{aligned}
$$

Define the combined unknown vector: $y = (x, s, \lambda, \mu)$ and consider the nonlinear system: $F(y) = 0$.

**IPM Iteration Steps.** A **primal-dual interior-point iteration** first take current iterate $y_k$ and linearize $F$ at $y_k$ (Newton step): $J_F(y_k) \Delta y = -F(y_k)$, then solve the linear system (this is the KKT matrix IPOPT factorizes), and then perform a line search or trust-region update until convergence: $y_{k+1} = y_k + \alpha \Delta y$ with the step length $\alpha$.

APPENDIX C: GPU ACCELERATION

Figure 9 highlights the significant benefit of GPU acceleration: lower bounds improve dramatically under batch-mode GPU execution.

APPENDIX D: A NAIVE EXAMPLE OF COMPLEMENTARITY-BASED REFORMULATION OF RELU NETWORKS

**Problem Formulation.** Consider a scalar input $x \in \mathbb{R}$ and a hidden layer with two ReLU neurons. Let $z^1 \in \mathbb{R}^2$ be the pre-activations and $\hat{z}^1 \in \mathbb{R}^2$ the post-ReLU activations:

$$
z^1 = W^1 x + b^1, \qquad \hat{z}^1 = \mathrm{ReLU}(z^1),
$$

with

$$
W^1 = \begin{bmatrix} 2 \\ -1 \end{bmatrix}, \quad b^1 = \begin{bmatrix} -1 \\ 0.5 \end{bmatrix}.
$$

The output layer is

$$
y = w^{2\top} \hat{z}^1 + b^2, \quad w^2 = \begin{bmatrix} 1 \\ -2 \end{bmatrix}, \quad b^2 = 0.1.
$$

We seek the smallest network output over an input interval: $\min_{x \in [-1,1]} y(x)$

**ReLU via Complementarity Constraints (NLP-CC).** As shown in Section 4.2, the ReLU function can be equivalently written as the solution of the convex quadratic program

$$\text{ReLU}(z) = \arg\min_{\hat{z} \geq 0} \tfrac{1}{2} \|\hat{z} - z\|_2^2,$$

whose optimality is characterized by the Karush–Kuhn–Tucker (KKT) conditions. For the $j$th neuron in layer 1, these conditions are

$$z_j^1 = \hat{z}_j^1 + t_j^1,$$
$$\hat{z}_j^1 \geq 0, \qquad t_j^1 \geq 0,$$
$$\hat{z}_j^1 \, t_j^1 = 0,$$

where $t_j^1$ is the dual variable associated with the nonnegativity constraint. These conditions are both necessary and sufficient and encode the identity $\hat{z}_j^1 = \max(0, z_j^1)$ exactly.

**Full NLP-CC formulation.** The resulting nonlinear program with complementarity constraints is

$$\min_{x, z^1, \hat{z}^1, t^1} \quad y = w^{2\top}\hat{z}^1 + b^2$$
$$\text{s.t.} \quad -1 \leq x \leq 1,$$
$$z^1 = W^1 x + b^1,$$
$$z_j^1 = \hat{z}_j^1 + t_j^1, \quad j = 1, 2,$$
$$\hat{z}_j^1 \geq 0, \; t_j^1 \geq 0, \quad j = 1, 2,$$
$$\hat{z}_j^1 \, t_j^1 = 0, \quad j = 1, 2.$$

This NLP-CC is exactly equivalent to minimizing the original 2-layer ReLU network over $x \in [-1, 1]$.

APPENDIX E: LINE-BY-LINE DESCRIPTION OF ALGORITHM 1

We now describe the SNOV algorithm in Algorithm 1 in detail.

**Line 1.** This line specifies the inputs and initializes the main parameters, including the optimality-gap tolerance $\epsilon$, the iteration counter $t$, and the waiting counter $\tau$.

**Line 2.** We compute the current lower bound $\underline{l}$ using $\alpha$-CROWN, and obtain the intermediate pre-activation bounds $[l^\ell, u^\ell]$ for all neurons, which will be reused for upper-bound computation. If $\underline{l} > 0$, the instance is already certified as safe and the algorithm terminates early.

**Line 3.** We compute the root upper bound $\bar{u}$ by solving the NLP with complementarity constraints (NLP–CC) using the pre-activation bounds $[l^\ell, u^\ell]$. If $\bar{u} < 0$, a counterexample is found and the model is declared unsafe.

**Line 4.** We initialize the domain set $\mathcal{D}$, which stores all active subdomains, together with their current bounds and feasible region $(\underline{l}, \bar{u}, \mathcal{C})$.

**Line 5.** We enter the branch-and-bound loop as long as the gap between the global lower and upper bounds exceeds $\epsilon$, the domain set $\mathcal{D}$ is nonempty, and the maximum iteration limit has not been reached.

**Line 6.** We compute the pattern-aligned strong-branching scores for unstable neurons and split on the neuron(s) with the highest score to generate new subdomains. We then select (pop) the subdomain $\mathcal{C}_i$ with the smallest lower bound from $\mathcal{D}$ for further bounding.

**Line 7.** We split the selected subdomain $\mathcal{C}_i$ into two child domains, corresponding to the active (0) and inactive (1) branches of the chosen ReLU.

**Line 8.** For each child subdomain $\mathcal{C}_i^{(h)}$, we compute a lower bound $\underline{l}(\mathcal{C}_i^{(h)})$ using $\beta$-CROWN and increment the waiting counter $\tau$ by one.

**Line 9.** We check whether $\tau$ exceeds the pre-defined limit $\tau_{\max}$.

**Line 10.** If the waiting limit is reached, we resolve the NLP–CC verification problem on $\mathcal{C}_i^{(h)}$ using the warm-started NLP solver (Sec. 4.2) to obtain the upper bound $\bar{u}(\mathcal{C}_i^{(h)})$, and reset $\tau$ to zero.

**Line 12.** We prune each subdomain if it either contains a counterexample ($\bar{u}(\mathcal{C}_i^{(h)}) < 0$) or is certified safe ($\underline{l}(\mathcal{C}_i^{(h)}) > 0$); otherwise, we keep it in $\mathcal{D}$ for future splitting.

**Line 14.** We update the global lower and upper bounds based on all remaining subdomains.

**Line 16.** Finally, we report the verification outcome: safe, unsafe, or inconclusive with the remaining gap between the global lower and upper bounds.

## B  REPRODUCIBILITY STATEMENT

All experiments use public datasets and open-source software (PyTorch, $\beta$-CROWN, Pyomo/IPOPT). We will release code, configs, and scripts to reproduce every table and figure, including fixed random seeds, exact perturbation radii $\delta$, tolerances $\epsilon$, and solver options. Hardware details (Apple M4 and an AMD 32-core workstation) are reported for reference, but results are reproducible on standard CPU/GPU machines.

## C  THE USE OF LARGE LANGUAGE MODELS (LLMs)

LLMs were used solely for grammar and wording; all technical content was independently verified by the authors for correctness and reliability.

