# OpenReview forum: "SNOV: A Scalable Near-global Optimal Verifier for   Neural Networks under Large Perturbations"
_ICLR.cc/2026/Conference — Submitted to ICLR 2026_

### Official Review · Reviewer_bdeZ · 2025-10-16

**Soundness:** 1
**Presentation:** 1
**Contribution:** 2
**Rating:** 2
**Confidence:** 4

**Summary:**

This manuscript introduces SNOV, a framework that combines $\beta$-CROWN lower bounds with a NLP solver for upper-bounds, all within the Branch-and-Bound (BaB) algorithm, enhanced by slope-guided warm starts and low-rank KKT updates. Some preliminary results related to verifications of neural networks are presented.

**Strengths:**

Although the BaB framework is well established for neural network verification, the authors aim to enrich it with engineering heuristics, such as slope-guided warm starts and low-rank KKT updates, that can reduce the overall computational cost.

**Weaknesses:**

At present, the paper does not convincingly showcase modeling capabilities on substantive application cases, and the theoretical developments seem largely incremental relative to known CROWN/BaB hybrids. The experimental section lacks the breadth and controls needed to support the efficiency claims (comprehensive baselines, ablations, and scaling studies). Clarity also suffers from repetition, ambiguous formulations, and language issues.

Substantial improvements could be made by addressing both the general comments and the specific remarks listed below.

### Major points

- Lack of consistency: there are some objects that are not denoted in the same way throughout the paper: feasible set is sometimes $\mathcal{X}$, sometimes $\mathcal{C}$, sometimes $\mathcal{B}$; the radius of the ball is $\gamma$, $r$, $\epsilon$ or $\epsilon_B$; the linear function of the last layer is $c$ or $\xi$. There’s inconsistent use of bold versus regular text. Some sentences are very informal, e.g., page 3, lines 149–150, or page 7, lines 360–362. Moreover, some objects or notations are used without being properly defined, e.g., per-neuron slopes, unstable fractions, strong-branching score, etc. Overall, the paper is extremely hard to read and follow, and presentation needs to be improved.

- Literature review is quite incomplete. Not all exact verification methods are MIP-based, and for example, SAT/SMT methods are never discussed. Similarly, approximate methods encompass a much larger number of works (abstract interpretation, zonotope domains, SDP relaxations) than the ones mentioned in Section 3.2.

- Experimental setup is not sufficient for deriving rigorous conclusions. Indeed, it is based on verifying **one single instance**, and the detailed network architecture—crucial for understanding the true problem size—is never mentioned. The paper also lacks a systematic analysis of how performance scales with perturbation magnitude.

- Computing exact optimal values $f^{*}$ with “exact or high-fidelity solvers” remains highly non-trivial in high-dimensional settings. No details are provided about this, despite this value being used to measure absolute and relative gaps.

### Minor points

- There should be a clear distinction between a given function, say $f$, and its evaluation at a particular point $x\in\mathbf{R}^n$, i.e., $f(x)$.
- If $\gamma$ is a real number, there is need to write things like $||\gamma||_{\infty}$. This is related to the lack of consistency point.
- The first part of Section 6 includes quite extensive solver-configuration details (e.g., sort_domain_interval=1) that are not essential for understanding the paper and should be moved to the Appendix. This would free space to describe the network architectures and core notions more clearly.

**Questions:**

1. Which exact $\beta$-CROWN variant is used? A short algorithmic description would help.

2. Why does splitting change the KKT system with rank at most $3$ in your formulation?

3. What is the exact "strong-branching score" used (formula, weights, normalization)?

4. In what sense exactly does the framework go beyond known BaB + LiRPA frameworks? Can you isolate a lemma or proposition that is new (not just a rephrasing) and explain its impact on tightness or complexity?

5. How do your results aggregate over larger sets (e.g., $100$–$1000$ test inputs)?

---

> ### Author Response · Authors · 2025-11-26
>
> **Thank you for the thoughtful feedback. We have revised the manuscript and respectfully request a second review.**
>
> **Weakness 1 (notation):**
>
> **Response:** We standardized notation (feasible set $\mathcal{C}$, radius $\delta$, final-layer linear map $c$); clarified bold vs regular symbols; and added precise definitions (e.g., strong-branching scores) in Section 2.
>
>
> **Weakness 2 (related work):**
>
> **Response:** Section 3.2 now surveys exact (MIP, SAT/SMT) and approximate methods, including abstract interpretation, zonotopes, SDP relaxations, and LiRPA-based verifiers, positioning our contributions within this landscape.
>
>
> **Weakness 3 (experiments):**
>
> **Response:** We verify hundreds of MNIST and CIFAR-10 instances and report full architectures and implementation details (Section 5, Appendix A). We analyze scalability vs $\delta$ and binaries (Tables 1–4, Figure 5) and the speed–accuracy tradeoff vs $\epsilon$ (Figure 7).
>
>
> **Weakness 4 (exact $f^\star$):**
>
> **Response:** We use MIP-derived $f^\star$ only when the solver returns certified optimality within the time limit; these settings are listed in Section 5. Cases without certified optimality are excluded from gap-to-$f^\star$ metrics; for harder cases we report certified bounds and speedups only.
>
> ## Minor Points
> **Minor points**:  There should be a clear ... If $\gamma$ is ... The first part ...
>
> **Response:** Notation now consistently distinguishes $f$ from $f(x)$. The scalar $\gamma$ issue is fixed. Non-essential solver flags were moved to Appendix A; Section 5 with Appendix A focuses on architectures and core methods.
>
>
> ## Questions
>
> **Q1 (beta-CROWN variant).**
>
> **Response:** We use standard beta-CROWN (node-wise linear relaxations with optimized $\beta$ including split constraints). A brief description is added to Section 2.
>
> **Q2 (why rank-3 updates).**
>
> **Response:** Splitting a neuron removes one complementarity constraint and adds up to two linear equalities, changing a small KKT block; hence a rank $\le 3$ update. Note that in the revised paper, we consider an extra constraint, resulting rank$\le 4$.
>
> **Q3 (strong-branching score).**
>
> **Response:**  Defined in Section 4.3: for each unstable neuron, score = predicted lower-bound improvement from approximate evaluation of both branches using fast beta-CROWN-style dual bounds; scores are normalized and pattern-aligned.
>
> **Q4($\epsilon$-optimality).**
>
> **Response:**  Our framework extends standard BaB+LiRPA methods in two main ways.
> (1) NLP--CC as a sound upper-bounding layer described in Sec.~4.1:
>
> **Proposition 1:** \[Sound upper bound from NLP--CC\]
> Consider a ReLU network with specification function $f$ and perturbation set $\mathcal{C}$, and let $f^*=\min_{x \in \mathcal{C}} f(x)$ be the global optimum of the original verification problem. Let $(x,\dots)$ be any feasible solution of the NLP with complementarity constraints (NLP--CC) obtained by our reformulation. Then $f(x) \ge f^\star$ i.e., every feasible NLP--CC solution yields a valid upper bound on $f^\star$.
>
> This holds because the NLP--CC is an exact reformulation of the original ReLU verification problem (same feasible set in $x$ and same objective), so any feasible NLP--CC solution provides a sound $\epsilon$-global upper bound, unlike PGD-style attacks that may miss positive optima. As shown in Sec. 5.1, SNOV$_u$ yields much tighter and more reliable upper bounds than PGD at a fraction of the cost of MIP. In addition, the low-rank KKT update mechanism in Sec. 4.2 accelerates NLP--CC after branching by enabling warm-started solves, giving a $2$--$5\times$ speedup per NLP call (Sec. 5.3).
>
>  (2) Pattern-aligned strong branching. In Sec. 4.3 we introduce a pattern-aligned strong branching strategy that reweights traditional filtered smart-branching scores using the NLP-derived activation pattern (active/inactive neurons at the incumbent), so that splits better aligned with this pattern are prioritized. As shown in Sec. 5.5, this yields more informative branches and systematically accelerates lower-bound tightening across branch rounds.
>
> **Q5.** How do your results aggregate ...
>
> **Response:** Our results now   aggregate over hundreds of test instances: Tables 1–4 report performance across multiple datasets and perturbation radii, covering 100  instances per setting.

---

> > ### Comment · Reviewer_bdeZ · 2025-11-28
> > **Response to Authors' rebuttal**
> >
> > I thank the authors for uploading the revised version.
> >
> > While it contains some improvements in the presentation of the main ideas, I believe the manuscript still does not meet ICLR standards. Numerous typos and writing ambiguities remain, some figures are not informative at all, others appear to be raw screenshots (with the mouse pointer visible), the networks used are still quite small, and the experimental section continues to lack important methodological and implementation details.
> >
> > Overall, my main concerns regarding the novelty of the approach and the clear demonstration of its potential have not been adequately addressed. Based on the revised version and the rather succinct responses provided to the reviewers’ comments, I prefer to maintain my initial score.

---

> > > ### Author Response · Authors · 2025-11-29
> > >
> > > **Response:**   We sincerely thank the reviewer for reading the revised manuscript and for the additional feedback. We acknowledge that some presentation issues remained and have now conducted another full proofreading pass to remove typos and ambiguities, and regenerated all figures as clean vector graphics (removing raw screenshots and improving readability). We have also restructured several paragraphs in Sections 3–4 to clarify the main ideas.
> > >
> > > Beyond presentation, we have added the **proposition 1** in Section 4.2 and made substantial technical and experimental revisions that directly address the concerns about scale, methodological clarity, and demonstrated potential. Major changes in the revised version include:
> > >
> > > #### 1. Substantially expanded experiments.
> > > We updated Tables 1–4 with results over 10–100 MNIST/CIFAR-10 instances under multiple perturbation radii, added PGD as a new baseline for upper bounds, and included additional figures illustrating (i) runtime scaling of SNOV$_u$ vs. MIP, (ii) warm-start speedups, (iii) overall BaB performance, and (iv) gains from the pattern-aligned branching strategy.
> > >
> > > #### 2. New pattern-aligned strong branching strategy.
> > > Section 4.3 introduces a pattern-aligned strong branching rule that leverages NLP-CC activation patterns, and Section 5.5 shows consistent improvements over filtered smart branching in terms of lower-bound tightening and BaB efficiency.
> > >
> > > #### 3. Clearer background and methodology.
> > > Section 2 now provides a concise introduction to NLP-CC, α/β-CROWN, branch-and-bound, and strong branching, with five new appendices that summarize the experimental details,   interior-point method,  GPU-acceleration,  an example of exact NLP-CC reformulation, and detailed description of Algorithm 1, so that readers from ML, formal methods, and optimization can follow the approach.
> > >
> > > #### 4. Improved related work and reproducibility.
> > > Section 3 has been rewritten to offer a more complete and balanced review of exact and approximate verification methods, and to more clearly position our contributions. We will also release code, models, and an extended library supporting multiple architectures (MLP, CNN, ResNet) to ensure reproducibility.
> > >
> > > Finally, we would like to emphasize that the core novelty is not only in combining existing components, but in (i) an exact NLP-CC upper-bounding procedure that yields significantly tighter bounds than gradient-based attacks (e.g., PGD), and (ii) a pattern-aligned branching strategy that uses NLP-induced activation patterns to accelerate BaB. We hope these clarifications and the strengthened empirical evidence help convey the contribution and potential of our approach, and we would be very grateful for the reviewer’s reconsideration.

---

### Official Review · Reviewer_xn2U · 2025-10-25

**Soundness:** 2
**Presentation:** 2
**Contribution:** 3
**Rating:** 2
**Confidence:** 4

**Summary:**

In the past years, neural network verifiers based on linear relaxations have been shown to be able to scale to verify large neural networks when combined with efficient branching heuristics. However, these methods struggle to provide tight certificates when perturbation radii are large. This work proposes a hybrid verification approach which combines convex-relaxation-based bound propagation methods to obtain lower bounds with more precise nonlinear-programming solvers for obtaining upper bounds. Both methods run in parallel and exchange information, enabling better branching decisions, fast warmstarts in the NLP solver and accelerated convergence to the true optima. The experimental evaluation demonstrates the capability of the proposed method which achieves a precision comparable to that of MIP solvers in significantly less time.

**Strengths:**

- The verification of neural networks is an important research topic
- Tackling the inability of bound-propagation-based verifiers to scale to large perturbation radii is a valuable contribution
- The experimental results are impressive, showing that the proposed method achieves MIP-like precision in significantly less time
- The authors introduce formulations which also enable the approach to be extended to transformers

**Weaknesses:**

- My biggest concern about the paper is the empirical evaluation. In neural network verification, we are generally interested in whether a network is robust or non-robust on a particular input for a given perturbation. There is little benefit in being able to obtain very precise bounds on the specification if loose bounds are already sufficient to prove that a property holds. Looking at the results, it seems that a number of experiments are done for cases where even the true lower bound is $<0$, this means that a counterexample exists. A cheap algorithm such as projected gradient descent could just be run to obtain counterexamples, this is done by any neural network verification tool before bound propagation is even started. Comparing the performance of different bound propagation methods in such cases doesn't make a lot of sense. I understand that PGD might not always find a counterexample, but with such large perturbations and lower bounds well below zero it should not be hard to find counterexamples. The only case where verification is actually possible (true lower bound $>0$) is that shown in Table 2. However, the cheap lower bounds obtained by $\alpha$-CROWN here are already sufficient to verify robustness, hence, SNOV does not provide any benefits in this case and is significantly slower than $\alpha$-CROWN and even slower than MIP. A proper evaluation should run SNOV against other state-of-the-art verifiers (such as GCP-CROWN [1]) on established benchmarks (such as those from [2]) and compare the verification time as well as the number of verified instances.
- Although the general concept of the SNOV verifier makes sense to me, there are a number of details on the algorithm (e.g. the rank-3 warm-starts) that are missing in the paper, therefore I am unable to assess whether the overall algorithm is correct. The authors also repeatedly mention that the NLP yields "incumbents and dual-like signals that drive strong branching" but the strong branching heuristic and the information flow from the NLP to $\beta$-CROWN is not explained in the paper.
- It is unclear how $\alpha$-CROWN and the other competing methods are evaluated. Is only a single bound propagation pass performed with these methods? If so, it is not surprising that they would produce looser bounds than the proposed method. SNOV should be evaluated against a complete verifier which performs branching as well (such as $\alpha, \beta$-CROWN and also GCP-CROWN which is enhanced with cutting planes). By running those for the same time budget as SNOV, a proper comparison between the methods would be possible.
- The runtime of the algorithm is significantly higher than that of other methods such as $\alpha$-CROWN and seems to explode when evaluated on slightly larger datasets such as CIFAR10 (see Table 3). I am therefore unsure whether the proposed method would actually scale to networks and datasets of practical size.
- The empirical evaluation lacks important details: Details on the neural network architectures that are evaluated are missing. The authors say that "We implement SNOV for MLP, CNN, ResNet, and Transformer architectures across three benchmark suites.", does this mean that all results in the paper are averages across these architectures? Separate results should be reported for these architectures. Besides this, the methods should be evaluated for multiple inputs and not only for one input as is currently done.
- Section 3.1 is quite imprecise and dense, therefore difficult to understand. E.g. what do the authors mean by "induce loose big-M" or "binaries explode"? This should be extended to provide more context and explain the points that are being made here in more detail.
- The section on "Low-rank KKT updates" (line 234ff) is dense and difficult to understand. A lot of the concepts being referred to here are not introduced.
- The algorithm introduces a number of hyperparameters ($\vartheta, \phi$) but there are no ablation studies and little justification for how these are selected.
- The authors state that some experiments are run on a small Mac machine while others are run on a server with 64 GPUs. It is unclear which experiments are run on which machine which is important to be able to assess the runtimes that are provided. Besides this, the type of GPUs and CPUs should be provided, and it should be clarified whether the approach runs on all 64 GPUs in parallel.
- The paper is full of typos, grammatical errors and incomplete sentences. I tried listing some of them below but eventually stopped taking note of all of them while reading. I would encourage the authors to thoroughly revise the paper with a focus on grammar and language.


### Minor weaknesses and typos
- The notation in the paper is somewhat chaotic and never properly introduced. The authors use $l, u$ as well as $\underline{l}, \overline{u}$ and $\underline{L}, \overline{U}$ to denote lower and upper bounds. E.g. in line 65, both are used in exactly the same context. The notation should be clarified and, if applicable, unified.
- Line 130: Shouldn't the objective function being minimised here be $f(x)$ which is previously introduced as the function representing the specification? $s(x)$ is never introduced
- Figure 3: What are $\xi(0)$ and $\xi(1)$? $\xi$ is introduced as the "specification coefficients" in line 122, but it's unclear what the indexing refers to here.
- Figure 6: The overapproximation area in the left part of the figure extends outside the blue bounds which is incorrect. The figure should be corrected so that the overapproximation is entirely contained in the lower/upper bounds.

- Line 121: is a task-specific specification **is** represented by function --> is a task-specific specification represented by **a** function
- Line 122: One of the common specification is --> One of the common specification**s** is
- Line 123: see Section for particular examples. --> Which section to the authors refer to here?
- Line 303: "ReLU can be written as the solution of the projection quadratic programming" --> this is not a sentence and I don't understand what it means. The authors should fix the grammatical errors here.
- Line 352: We observe that bound propagation based methods efficiently produce the bounds but far loose --> We observe that bound propagation based methods efficiently produce **bounds, but that they are far too loose**
- Line 353: The relative relaxation gap $\overline{\Delta}_{0.1}$ are between --> The relative relaxation gap $ \overline{\Delta} _{0.1} $ **is** between
- Line 353: "Table 1 and Table 2." is not a valid sentence, this needs to be rewritten.
- Line 355: Section5 --> Section 5
- Table 1 shows that NLP solver reaches up to 7 times faster than MIP --> Table 1 shows that **the** NLP solver reaches up to 7 times faster than MIP. Also what is reached here? The sentence makes no sense in its current form
- Line 361: making the MIP **is** faster than NLP --> making the MIP faster than NLP
- Line 364: algorithm taking advantages of --> algorithm taking **advantage** of
- Line 365: the efficiency of bound propagation method --> the efficiency of **the** bound propagation method
- Table 1 and all other tables: "when large perturbations" should be replaced with "**under** large perturbations". Also "Verifying One Image of MNIST Dataset" should be "Verifying One Image of **the** MNIST Dataset"
- Line 401: with the lower bounds of α−CROWN method --> with the lower bounds of **the** α−CROWN method
- Line 403: to take **a** hundreds of seconds --> to take hundreds of seconds
- Line 404: the proper initialization of NLP solver --> the proper initialization of **the** NLP solver
- Line 405: the efficiency of NLP has about 18 times improvement --> the efficiency of NLP **improves by a factor of 18$**
- Line 407: As shown in Section 5 **that** bound propagation method like α−CROWN is sensitive to large perturbations --> As shown in Section 5, bound propagation method**s** like α−CROWN **are** sensitive to large perturbations,
- Line 408: Grammar, producing a too loose lower bounds --> producing loose lower bounds

### References

[1] Zhang, H., Wang, S., Xu, K., Li, L., Li, B., Jana, S., Hsieh, C.-J. & Kolter, J.Z. (2022) General Cutting Planes for Bound-Propagation-Based Neural Network Verification. doi:10.48550/arXiv.2208.05740.

[2] Brix, C., Bak, S., Johnson, T.T. & Wu, H. (2024) The Fifth International Verification of Neural Networks Competition (VNN-COMP 2024): Summary and Results. doi:10.48550/arXiv.2412.19985.

**Questions:**

- How are the competing bound propagation methods run? Is branching conducted for these?
- How were the hyperparameters selected?
- Which experiments are run on which of the machines and are all of the GPUs used in parallel?
- The authors claim to solve the problem to $\epsilon$-optimality but then employ a heuristic stopping criterion which, as far as I see, does not guarantee $\epsilon$-optimality. Could the authors clarify what exactly they mean when describing the optimality here?

---

> ### Author Response · Authors · 2025-11-26
>
> **Thank you for the thoughtful feedback. We have thoroughly revised the manuscript and respectfully request a second review.**
>
> **Weakness 1:**  Scope and necessity of tight bounds.
>
> **Response:** We agree that when loose bounds already certify robustness or quickly find counterexamples, tighter bounds are unnecessary. Our goal is broader: SNOV efficiently estimates an $\varepsilon$–global optimum $f^\star$, yielding a quantitative safety margin and a small interval containing $f^\star$, which is especially valuable when bound propagation or PGD are inconclusive under limited compute. On the first 100 CIFAR-10 cases with $\delta=0.03$ (42 with $f^\star<0$, 58 with $f^\star>0$), SNOV either early-stops (once safety/violation is decided) or terminates when the gap drops below $\varepsilon$. Figure 5 (Sec. 5.2) shows MIP runtime growing nearly exponentially with the number of binaries, while SNOV$_u$ remains stable and gains a large advantage beyond $\sim$100 binaries. Finally, many  state of the art verifiers (e.g., $\alpha$–$\beta$-CROWN, GCP-CROWN) use BaB+$\beta$-CROWN; our contribution is a stronger upper-bounding module and a pattern-aligned branching strategy, yielding tighter upper bounds than PGD at a fraction of MIP’s cost and faster lower-bound tightening (Sec. 5.1, Sec. 5.5).
>
> **Weakness 2:** Method clarity.
>
> **Response:** Section 4.2 details the rank-3 warm start and its use in Algorithm 1. Section 4.3 defines pattern-aligned strong branching, scoring neurons by alignment with the NLP-derived incumbent; Section 5.5 validates the impact with beta-CROWN.
>
>
> **Weakness 3:** Evaluation setup.
>
> **Response:** IBP, CROWN-IBP, CROWN, and $\alpha$-CROWN run once (no BaB) and are compared to stand-alone SNOV$_u$ to isolate upper-bound tightness (Tables 1–4). For complete BaB, since alpha–beta-CROWN, GCP-CROWN, and SNOV share beta-CROWN lower bounds, we compare the differing parts: our NLP upper bounds vs PGD (Section 5.1) and our pattern-aligned branching (Section 5.5).
>
>
> **Weakness 4:** Runtime.
>
> **Response:**  SNOV$_u$ calls an NLP (KKT reformulation), so per-call cost exceeds single-pass bound propagation, but scaling is far milder than MIP. On 100 CIFAR-10 cases with $\delta=0.03$, SNOV$_u$ finishes in at most 30 s, while MIP grows from seconds to thousands as binaries increase (Figure 5, Section 5.2). Low-rank warm start gives up to $5\times$ speedup after branching; GPU-accelerated lower bounds further reduce end-to-end time.
>
> **Weakness 5:** Experimental clarity.
>
> **Response:** Appendix A now provides full architectural and training details. Results are for two MLPs (small and larger). We also evaluate on the first 100 CIFAR-10 inputs.
>
> **Weakness 6:** Related work precision.
>
> **Response:** Section 3.1 is rewritten around complete vs incomplete verifiers with clearer terminology and positioning.
>
> **Weakness 7:** Low-rank KKT exposition.
>
> **Response:** The section is simplified; a self-contained IPM primer is added in Appendix B.
>
>
> **Weakness 8:** Hyperparameters.
>
> **Response:** We removed the slope-guided initialization and its hyperparameters $(\vartheta,\phi)$. We analyze $\delta$, the stopping tolerance $\epsilon$, and the branching weight $\lambda$ (Tables 3–4; Figures 7–8).
>
> **Weakness 9:** Hardware clarity.
>
> **Response:** We specify hardware per experiment: upper-bounding results on a single Mac (Apple M4 CPU/GPU) (Section 5.1-5.3); BaB experiments on a single server GPU (Section 5.4-5.4). No 64-GPU parallelism is used.
>
>
> **Weakness 10:** Writing quality.
>
> **Response:** Grammar and typos are corrected throughout; changes are highlighted.
>
> ### Minor weaknesses and typos
> **Minor weaknesses and typos (11–33):**
>
> **Response:** All corrected.
>
> ### Questions:
> **Q1:** Competing methods.
>
> **Response:** IBP, CROWN-IBP, CROWN, and $\alpha$-CROWN are run in standard single-pass mode to measure one-shot bound tightness and cost. Branching is only used in SNOV with beta-CROWN.
>
>
> **Q2:** Hyperparameters.
>
> **Response:** Refer to the response to weakness 8.
>
> **Q3:** Hardware.
>
> **Response:** Refer to the response to weakness 9.
>
> **Q4:** $\epsilon$-optimality.
>
> **Response:** Guaranteed only when early stopping is disabled or not triggered and the BaB gap satisfies $\bar u - \underline \ell \le \varepsilon$. Early stopping (if $\bar u < 0$ or $\underline \ell > 0$) is a practical speed heuristic; in that mode we do not claim $\epsilon$-optimality.

---

### Official Review · Reviewer_TAhY · 2025-10-27

**Soundness:** 1
**Presentation:** 1
**Contribution:** 2
**Rating:** 2
**Confidence:** 4

**Summary:**

The paper presents a novel neural network verifier, named SNOV, that combines state-of-the-art branch-and-bound based on linear relaxations with the use of of a non-linear-programming solver (NLP) to compute upper bounds, as opposed to the customary use of local grandient-based optimizers and primals from the lower bounding algorithms.
This is aimed at what the authors call the "large-perturbation regime".
Experimental results show that SNOV is significantly faster than the employed MILP solver on the considered benchmarks.

**Strengths:**

The idea to use NLP for BaB upper bounds is novel and potentially very interesting for the neural network verification community.
It does seem to be significantly faster than the considered MILP solver on the provided benchmarks.

**Weaknesses:**

What I believe to be the main weaknesses of the paper are related to the presentation and the experimental section.

**Presentation**.

The presentation assumes familiarity with concepts related to interior-point methods. I do not think this is reasonable, and the authors should provide at least a basic introduction in the appendix. KKT conditions are mentioned (and are related to some of the technical improvements) without any technical explanation. Given that neural network verification is a mixed community, with people coming from ML, from formal methods, and from optimization, this appears to be particularly necessary.

Furthermore, important details are omitted: a new branching strategy is introduced, but no details are presented. It is unclear to me what networks are employed for the experiments. It is also not quite clear what is the precise purpose of the complementarity reformulation.

Additionally, I would encourage the authors to tone down the narrative a bit. For instance, it's hard to say that the proposed algorithm has "consistent improvements in scalability and reliability over six state-of-the-art baselines". Improvements are in either scalability or accuracy, and in a very limited experimental setup (see below), and most of these baselines are really far from the state-of-the-art for neural network verification (IBP has never been, for instance, it is mostly used for training purposes).

**Experiments**.

The experiments appear to be carried out over extremely small networks (judging from the MILP runtimes), and over relatively small perturbation radii (it is common to use up to $\epsilon=0.3$ and $\epsilon=8/255$ for MNIST and CIFAR-10, respectively).
Furthermore, no branch-and-bound baseline is provided.
Speed improvements over MILP solvers are not surprising of their own, and provide no indication on whether the proposed approach would be beneficial to the state-of-the-art (for instance, whether it speeds up alpha-beta-CROWN).
The fact that no code is provided makes it even harder to assess the experimental results.


Given the inconclusive experimental section and the rushed presentation, I do not believe the paper is ready for publication at this stage. But I would be happy to increase my score if these concerns are addressed.

**Questions:**

- Are you running the NLP sequentially over each branch-and-bound subdomain? If so, is the point associated to the current subdomain  upper bound feasible for it (in other words, does it satisfy all split constraints)? How can this scale?
- How are intermediate pre-activation bounds set for the MILP? MILP solvers appear to benefit from very tight bounds (see discussion in the GCP-CROWN paper).
- Can you include other scalable BaB methods (the standard alpha-beta-CROWN, at the very least) among the baselines?
- Could you please comment on "A Branch and Bound Framework for Stronger Adversarial Attacks of ReLU Networks", Zhang et al., ICML22? This work focuses on improving the upper bounding part within branch and bound, and it seems extremely relevant for the submission. It would be very important to also benchmark against it.

---

> ### Author Response · Authors · 2025-11-26
>
> **Thank you for the thoughtful feedback. We have thoroughly revised the manuscript and respectfully request a second review.**
>
> **Weakness 1:** Presentation. ...
>
> **Response:**  We have added a detailed introduction to interior-point methods (IPMs) and KKT conditions in Appendix~B, explaining how IPMs solve the NLP-CC by forming the KKT system with barrier terms for inequality constraints and then applying Newton steps to a sequence of smoothed KKT systems until barrier terms approaching zeros.
>
> **Weakness 2:** Furthermore, important details are omitted: ...
>
> **Response:** We have described the pattern-aligned branching strategy in detail in     Section 2 (conceptual background) and Section 4.3 (methodology), with experimental validation in Section~5.5. The network architectures and additional experimental settings are provided in Appendix A. We also clarify the complementarity reformulation in Section 2 and explain its role in Section 4.2: it gives an exact encoding of ReLU, ensuring that the NLP solution is a valid upper bound for the original verification problem.
>
>
>
> **Weakness 3:** Additionally, I would encourage the authors ...
>
> **Response:** We have toned down the narrative, clarified that improvements are limited to our experimental setting, and repositioned the compared methods rather than calling them “state-of-the-art.” We also report additional comparisons (PGD, MIP, and FSB for branching) to better contextualize our results.
>
> **Weakness 4:** Experiments. ...
>
> **Response:** We now clarify that we use two networks (100 and 522 neurons) with perturbation radii from 0.01 to 0.3  ($\approx$ 8/255) and report updated results in Tables 1–5 and Appendix A. We focus on settings where MIP can still solve to (near) optimality so that optimality gaps are measurable; larger networks where MIP times out are left for future work. We also add a BaB baseline: our pattern-aligned strong branching improves  $\beta$-CROWN in Section~5.5 and is in principle applicable to  $\alpha-\beta$-CROWN. The code has been submitted and will be released on GitHub upon publication.
>
>
>
> **Weakness 5:** Given the inconclusive experimental ...
>
> **Response:** Thank you for the feedback. We have substantially revised the manuscript, clarified the presentation, and added extensive new experiments (summarized at the beginning of the paper) to address these concerns.
> ### Questions:
> **Q1.** Are you running ...
>
> **Response:** We do **not** solve a full NLP on every BaB subdomain. When we do call NLP on a node, we update the constraints in the NLP-CC formulation to match the split, so the resulting solution is feasible for that subdomain and gives a valid upper bound. In practice, we (i) invoke NLP only intermittently (determined by $\tau_{\max}$ in Section 4.4) when lower bounds have improved sufficiently, rather than at every node, (ii) use NLP purely as an upper-bounding oracle (so many nodes never require an NLP solve), and (iii) warm-start with low-rank KKT updates (Sec.~4.2), which yields up to about 5× speedup. This selective, warm-started use makes the approach scalable.
>
> **Q2.** How are intermediate pre-activation...
>
> **Response:** We first compute tight intermediate pre-activation bounds with $\alpha$-CROWN and then pass these bounds to the MILP solver.
>
> **Q3.** Can you include ...
>
> **Response:** Since both  $\alpha-\beta$-CROWN and SNOV are BaB methods that use $\beta$-CROWN for lower bounds, we focus on the differing components: upper bounding and branching. We compare our upper-bounding method against the PGD-based upper bounds used in $\alpha-\beta$-CROWN (Sec. 5.1, Table~1 in this file), where our approach yields tighter and more reliable upper bounds, and we evaluate our pattern-aligned strong branching as an improvement to $\beta$-CROWN in Sec. 5.5.
>
> **Q4.** Could you please comment ...
>
> **Response:** We now cite and discuss Zhang et al. (ICML’22) and emphasize that their framework also improves the upper-bounding component in BaB, primarily using PGD-based attacks. In contrast, SNOV$_u$ uses an NLP-based upper bound that is more precise and reliable (never failing even when the optimum is positive), at modest extra cost but still far faster than MIP. See Sec.~5.1. Other BaB components are largely compatible and could be transferred between the two frameworks.

---

### Official Review · Reviewer_Lits · 2025-10-29

**Soundness:** 2
**Presentation:** 1
**Contribution:** 2
**Rating:** 2
**Confidence:** 5

**Summary:**

This paper presents SNOV, an exact neural network verifier that combines NLP-based primal heuristics with dual bounds obtained from $\alpha$-$\beta$-CROWN.

Overall, the paper presents several interesting ideas, but I find the presentation to be lacking several important mathematical details, especially regarding the parts that appear to be new in the paper. Furthermore, numerical results are incomplete / hard to follow, and I found some flaws in the experimental setting / solver comparison. My score is largely motivated by the fact that key information is missing from the paper, and the limitations of the numerical experiments. I believe the core ideas have some merit but the paper needs a significant re-write before being ready for publication.

**Strengths:**

* The paper considers large input-domain perturbations, which is less commonly tackled in existing verification literature
* The paper leverages primal and dual information to accelerate the efficiency of NLP primal search and dual $\alpha$-$\beta$-CROWN bound propagation
* The idea of low-rank KKT updates following branching on a neuron activation is new, however its presentation could have been more exhaustive

**Weaknesses:**

* I do not consider the "hybrid scheme" (Section 3.2) of SNOV to be a strong novelty. As noted in the introduction, existing verifiers combine a primal (to find adversarial examples if they exist) and a dual component (to obtain certified bounds). For instance, $\alpha-\beta$-CROWN implements primal heuristics like gradient-based attacks.
* Section 3 partitions existing works into "exact" MIP-based vs "approximate" branch-and-bound methods; I do not agree with the paper's classification.
  * The standard terminology in the NN verification literature is to distinguish between
    * _complete_ verifiers: methods that are guaranteed to either certify robustness or find an adversarial example given sufficient time. Virtually all such methods are based on a mixed-integer representation of the trained neural network, input domain and verification property; methods and tools differ in how they evaluate primal/dual bounds and in their implementations. $\alpha$-$\beta$-CROWN, Marabou, nnenum, CORA, etc... are complete verifiers.
    * _incomplete_ verifiers: methods that may terminate without a definitive answer, i.e., which are heuristic in nature. CROWN, $\alpha$-CROWN or gradient-based attacks are incomplete. Note that complete verifiers often combine a incomplete verifier with a branch-and-bound scheme.
  * the paper's classification between Mixed-Integer Programming and Branch-and-Bound does not capture the fact that i) MIP-based methods rely on branch and bound for completeness and ii) the methods cited in Section 3.2 are based on MIP formulations.

* Building on the above comment, Section 3 would benefit from a re-organization, and deserves additional relevant references such as i) existing verifiers such as CORA, Marabou, nneum, etc.. and ii) existing works that propose primal algorithms, eg.:
  * [_Optimization Over Trained Neural Networks: Taking a Relaxing Walk_](https://arxiv.org/abs/2401.03451)
  * [_Nonlinear Optimization with GPU-Accelerated Neural Network Constraints_](https://arxiv.org/abs/2509.22462)

* Several methodological components are mentioned but not explained, e.g. the paper makes several mentions of "strong branching scores" but these do not appear to be described anywhere in the paper
* Tables 1-7 have inconsistent notations: some use $|\gamma|_{\infty}$, some use $\gamma$.
* The use of $\gamma$ notation also conflicts with the notation $\epsilon_{B}$ used to define the input domain at the beginning of Section 6. * In Tables 4 & 5, subscript $u$, $u_{ini}$ and $u_{adj}$ are not defined.
* Table 3 and Table 6 are identical
* When solving verification tasks, one can terminate the solve as soon as an adversarial example is found or the instance is proven to be robust. Adding this termination criterion would likely affect the performance results reported in Section 6
* Several claims, e.g., lower number of branch-and-bound nodes, are not supported by any results
* Parts of the text are not proper English sentences, e.g. "Table 1 and Table 2." (l. 353)

**Questions:**

* Can the authors comment on the choice of representing ReLU activation using complementarity constraints as opposed to simply representing them as nonlinear functions in the NLP formulation?
* How would the complementarity constraint approach handle non-piecewise activation functions, e.g., sigmoid?
* Section 4 makes several mentions of $\beta$-CROWN as the method used to obtain node-level lower bounds. Should this have been $\alpha$-CROWN instead? Several parts of the text refer to algorithmic components of $\alpha$-CROWN.

---

> ### Author Response · Authors · 2025-11-26
>
> **Thank you for the thoughtful feedback. We have thoroughly revised the manuscript and respectfully request a second review.**
>
> **Weakness 1:** I do not consider the "hybrid scheme"...
>
> **Response:** We agree that combining primal and dual components is standard. We compute a principled upper bound via a nonlinear program with complementarity constraints and pair it with a relaxation-based lower bound, yielding a hybrid verifier that contracts the gap to convergence. Our key contribution is a **bidirectional** efficiency link between the primal (upper-bounding) and dual (lower-bounding) solvers. On 100 CIFAR-10 cases, SNOV$_u$ produces upper bounds up to two orders of magnitude tighter than PGD, while closely matching exact MIP results:
> | Metric                  | PGD   | SNOV$_u$    |
> | ----------------------- | ----- | --------- |
> | $\Delta_{0.03}$         | 0.204 | **0.003** |
> | $\bar{\Delta}_{0.03}$   | 5.781 | **0.008** |
> | $\Delta_{0.01}$         | 0.17 | **0.0005** |
> | $\bar{\Delta}_{0.01}$   | 0.32 | **0.0009** |
>
>
>
> **Weakness 2:** Section 3 partitions existing works ...
>
> **Response:** We  agree that the complete vs.  incomplete distinction is more standard and precise than our original “MIP vs.\ BaB” categorization. Accordingly, we have revised Section 3 to organize related work in terms of complete and incomplete verifiers, and to explicitly note that MIP-based methods are themselves implemented via branch-and-bound and that the methods in Section 3.2 are based on MIP formulations.
>
> **Weakness 3:** Building on the above comment, ...
>
> **Response:** Thanks. We re-organized Section 3 and commented on more comprehensive literature (including the suggested ones) to significantly improve the paper.
>
> **Weakness 4:** Several methodological components are ...
>
> **Response:** We have added a detailed description of the strong-branching scores and our pattern-aligned branching strategy in Section 2 (conceptual background) and Section 4.3 (methodology) of the revised paper, and we provide experimental validation in Section~5.5.
>
> **Weakness 5:** Tables 1-7 ...
>
> **Response:** Corrected with   $\|\delta\|_{\infty}$ consistently.
>
> **Weakness 6:** The use of $\gamma$ notation...
>
> **Response:** We have corrected the notation by using  $\delta$ consistently to denote the input perturbation and removed the conflicting use of  $\gamma$ and  $\epsilon_{B}$ and defined all used symbols.
>
> **Weakness 7:** Table 3 and Table 6 ...
>
> **Response:** We have renewed the results in these tables with statistical performance compared with more baselines rather than just simple case test. See Table 1-4.
>
> **Weakness 8:** When solving verification ...
>
> **Response:** We provide a user-selectable option to either terminate early or continue to close the optimality gap; this setting is documented in Section~6 and reflected in the reported performance.
>
> **Weakness 9:** Several claims, e.g., ...
>
> **Response:** We have removed unsupported claims and, where appropriate, added corresponding validation experiments to substantiate the remaining statements.
>
> **Weakness 10:** Parts of the text are not proper English sentences, ...
>
> **Response:** Improved the entire manuscript to ensure complete sentences.
>
>
>
> ## Questions:
> **Q1.** Can the authors comment on the choice of representing ReLU activation ...
>
> **Response:** Our choice is mainly motivated by two factors.
> (1) The complementarity-constraint (CC) formulation is an exact encoding of ReLU, while smooth surrogates (e.g., softplus) only approximate ReLU and can break the guarantee that the NLP optimum is a valid upper bound on the true ReLU problem.
> (2) The CC formulation yields a well-structured NLP with bounded, well-behaved derivatives in our smoothed KKT condition-based reformulation, which off-the-shelf solvers (e.g., IPOPT) can handle efficiently in practice. we clarify this distinction in Section~2.
>
>
>
> **Q2.** How would the complementarity constraint approach handle non-piecewise activation functions, e.g., sigmoid?
>
>
> **Response:** For smooth, non–piecewise activations such as sigmoid, we simply model them directly as standard nonlinear functions in the NLP, since they are already differentiable and bounded and do not require any complementarity reformulation. The complementarity-constraint approach is mainly needed for non-smooth piecewise activations (e.g., ReLU/Heaviside), where it provides an exact encoding.
>
>
> **Q3.** Section 4 makes several mentions...
>
>
> **Response:** We intentionally use both $\alpha$-CROWN and $\beta$-CROWN. $\alpha$-CROWN is used once at the root to compute intermediate bounds for all unstable neurons before any splitting. After branching starts,  $\beta$-CROWN is employed to compute node-wise lower bounds that explicitly incorporate the split constraints within the BaB tree. We have clarified this division of roles and the corresponding bounding strategies in Section 2 (background) and Section 4 (methodology).

---

### Author Response · Authors · 2025-11-26
**Overview and Main Changes**

We thank all reviewers for their thoughtful and constructive feedback. We have
**thoroughly revised the paper** (changes highlighted in red) and summarize the main updates below:


1. **[Key improvement 1: Expanded and strengthened experiments.].** We updated Tables 1–4 with results averaged over 10 or 100 MNIST and CIFAR-10 instances under multiple perturbation radii, demonstrating the tightness and robustness of our upper-bounding method. We added Projected Gradient Descent (PGD) as an additional baseline to contrast its behavior with SNOV$_u$ in terms of bound tightness and upper-bounding success rate. We also added Figures 5–8 to illustrate: (i) the runtime scaling of SNOV$_u$ versus MIP, (ii) the speedup from the proposed warm-start strategy, (iii) the overall performance of SNOV within BaB, and (iv) the improvement in lower bounds from the pattern-aligned strong branching strategy.
2. **[Key improvement 2: New branching strategy.].**
 We introduce a **pattern-aligned strong branching** strategy in Section 4.3 and compare it against the traditional filtered smart branching in Section~5.5, showing consistently faster lower-bound tightening and improved BaB efficiency.

3. **[Key improvement 3: Added background on core concepts.].** We now provide a concise background in Section 2 on nonlinear programming with complementarity constraints (NLP--CC), bound-propagation methods ($\alpha$-CROWN, $\beta$-CROWN), branch-and-bound, and strong branching, so that readers from ML, formal methods, and optimization can follow the methodology.
4. **[Key improvement 4: More complete literature review and released code.].** Section~3 has been rewritten to offer a more comprehensive review of exact and approximate verification methods, clearly positioning our contributions. We also submit code and datasets, including an extended library supporting multiple architectures (MLP, CNN, ResNet), to facilitate reproducibility and broader use.
We address each reviewer’s comments in detail below.

---

### Meta-Review · Area_Chair_8rx6 · 2026-01-07

**Summary:**

The reviewers agree that the paper lacks proper numerical evaluation of the proposed method. Namely, in neural network verification, we are generally interested in whether a network is robust or non-robust on a particular input for a given perturbation. A number of experiments are done for cases where even the true lower bound is <0, which means that a counterexample exists. Such a counterexample can be found using efficient approximate methods, making the numerical results not useful.

**Reviewer Concerns:**

Lack of consistency in the writing adds to this weakness, which has been addressed in revision.
The weaknesses in experimental set-up has not been addressed fully.

**Reviewer Scores:**

I believe they would have remained the same.

---

### Decision · Program_Chairs · 2026-01-26

Reject